# Anti-Obesity Effects of *Ecklonia cava* Extract in High-Fat Diet-Induced Obese Rats

**DOI:** 10.3390/antiox11020310

**Published:** 2022-02-03

**Authors:** Muhammad Aleem Abbas, Naila Boby, Eon-Bee Lee, Joo-Heon Hong, Seung-Chun Park

**Affiliations:** 1Laboratory of Veterinary Pharmacokinetics and Pharmacodynamics, Cardiovascular Institute, College of Veterinary Medicine, Kyungpook National University, Daegu 41566, Gyeongsangbuk-do, Korea; syedaleemabbas77@knu.ac.kr (M.A.A.); nailaboby@korea.kr (N.B.); eonbee@gmail.com (E.-B.L.); 2Bacterial Disease Division, Animal and Plant Quarantine Agency, 177 Hyeksin 8-ro, Gimcheon-si 39660, Gyeongsangbuk-do, Korea; 3Department of Food Science and Technology, Daegu Catholic University, Hayang-ro 13-13, Gyeongsan-si 38430, Gyeongsangbuk-do, Korea

**Keywords:** anti-obesity, *Ecklonia cava*, brown alga, high-fat diet, 3T3-L1, leptin and ghrelin, liver biomarkers, phlorotannin, obesity, ROC, antioxidant

## Abstract

Obesity is becoming a global epidemic as a result of high-calorie food intake and unhealthy lifestyles. Different marine plants, especially brown algae (*Ecklonia cava*), are traditionally used to treat different health-related issues. The study was carried out to investigate the anti-obesity properties of *E. cava* 70% ethanol extract. To evaluate the anti-obesity effect of *E. cava*, both in vitro and in vivo tests were performed. *E. cava* suppresses pre-adipocyte 3T3-L1 differentiation in a dose-dependent manner. In HFD-induced obese rats’ models, administration of *E. cava* 125, 250, and 500 mg/kg significantly decreases total body weight and organs, especially liver weight, in all treatment groups. Adipose tissue weight, including subcutaneous, epididymal, peritoneal, and mesenteric adipose tissue, was markedly reduced in *E. cava*-treated HFD rats in dose-dependent manners. In addition, liver-related biomarkers AST, ALP, ALT, and GGT were evaluated; the lower level of liver-related biomarkers indicates no liver injury or fatty liver issue in *E. cava* HFD treatment groups. In addition, *E. cava* treatment has significant effects on the expression of adipogenic and lipogenic (PPAR-γ, FAS, LPL, and SREBP-1c) genes. Altogether, these results show the anti-obesity effect of *E. cava*. We concluded that *E. cava* could be a potential candidate for the prevention of obesity-induced by a high-fat diet.

## 1. Introduction

Obesity is increasing all over the world and is causing genuine medical problems. The horrible fact is that the number of obese patients is increasing day by day. It is affecting lives without discrimination of gender, age, and species [1]. According to the world health organization (WHO), the number of people suffering from obesity has almost tripled since 1975. Recent studies show that 39% of the adult population is overweight and 13% is obese. A pooling analysis data show that a total of 2.16 billion will become overweight and 1.12 billion individuals will be obese in 2030 [2,3]. However, the recent COVID-19 pandemic situation makes it worse; because of strict restrictions on movement and limited access to outdoor events, the number of obese patients will increase all over the world [4]. A Korean research study was performed on school-going children’s body weight, which showed that 5.80% body weight increased during this pandemic time. Other factors such as cholesterol, low-density lipoprotein (LDL), high-density lipoprotein (HDL), and triglycerides (TG) were also disturbed [5]. However, as compared to the rest of the world, the number of obese persons is very low in South Korea, but the prediction statistics show that it will increase in upcoming years [6]. The major reason behind obesity is energy imbalance because of the absence of activity and inordinate food intake. This disease can lead to various pathological conditions such as nonalcoholic fatty liver disease (NAFLD), cardiovascular disease (hypertension), arteriosclerosis, diabetes, respiratory complications, osteoarthritis, and certain forms of cancer [1,7,8]. Consequently, obesity can decrease the life span by impairing the quality of life. Thus, the limiting rate of obesity could not only reduce the risk of mortality and morbidity per se but also the obesity-associated diseases [9,10]. Although diet control and exercise can effectively reduce obesity, but treatment using therapeutic agents along with exercise and controlled diet makes this process more effective. Currently, there are several therapeutic agents available for the treatment of obesity, including orlistat, sibutramine, diethylpropion, and phentermine [11]. However, due to the unfavorable impacts of these drugs, such as cardiovascular sickness, overstimulation, and abuse, other strategies, such as dietary supplements and herbal products, are required as an alternative therapy. Prevention by dietary changes is the best choice as compared to anti-obesity medicine or surgery. As a result, there is a need for new weight loss compounds that are both efficient and safe. Natural products are considered safe and have the potential to treat different medical issues. Natural phenolic compounds need more attention because some may be used as an alternative for the development of anti-obesity approaches [12].

Many natural substances, including marine algae, have a regulatory effect on lipid metabolism. Brown algae have long been used in Asian diets and in folk medicine. *Ecklonia cava* (*E. cava*) is an edible brown alga. It is mostly found in subtidal areas of Korea and Japan. *E. cava*’s health effects and chemistry have been researched, contributing to its use in nutritional supplements and herbal remedies [13]. This alga contains unique polyphenols called phlorotannin, and *E. cava* is enriched with these phlorotannins as compared to other brown algae. Several in vitro and in vivo experiments have shown that *E. cava* phlorotannins have possible health benefits. The phlorotannin’s dieckol, eckol, 6, 60-bieckol, phlorofucofuroeckol A, 7-phloroeckol, fucodiphloroethol G, and dioxinodehydroeckol can be isolated from *E. cava* and have a wide range of biological activates. Previous research studies show that these phlorotannins have antioxidant, anti-inflammatory, anticancer, antidiabetic, antihypertensive, radioprotective, anti-HIV, and anti-allergic properties [14,15,16]. A dieckol-rich concentrate of *E. cava* improves lipids and glucose metabolism in C5&BL/KsJ-db/db (db/db) mice model of type 2 diabetes [17]. In mice, dieckol and seapolynol isolated from *E. cava* have an antihyperlipidemic effect [18]. *E. cava* is also used as herbal medicine in the form of an extract called *Seanol*, which is high in phlorotannin [19]. Research studies also show that the use of *E. cava* as a daily food supplement is safe for humans over 12 years of age. Furthermore, there are no side effects on animal health [20].

Additionally, the research literature shows that *E. cava* has a different activity that can predict its anti-obesity effects. *E. cava* extract has been reported to have antilipidemic and anti-obesity related effects both in vitro and in vivo study; most of the in vitro studies were performed on 3T3-L1 cells, and in vivo studies were performed on mice. Previous in vivo studies missed important factors like hormones, obesity-related biomarkers, and important liver biomarkers, which are related to obesity. There is a lack of well-performed studies related to *E. cava* efficacy for high-fat diet-induced obesity in the scientific literature. However, in any case, the effects of *E. cava* supplemented high-fat diet-induced obesity have not been specifically studied in rats. This *Ecklonia cava* anti-obesity research was conducted for the first time on Sprague–Dawley rats. It is advantageous to use rat so that we can see the change in body weight as an experimental model. Furthermore, the reduction in body weight was analyzed with receiver operating characteristic standards, as well as the major organs like liver, kidney, and spleen were observed, and weights were recorded. The histopathological analyses were performed in this study, the samples taken from liver and fats. The most important factors related to obesity, namely, HDL, LDL, total cholesterol, total glycerides, insulin, glucose, and gastric inhibitory peptides (GIP) levels, were analyzed in this study. Leptin and ghrelin hormones, which are directly related to hunger and feeling of fullness, were checked. To find the effects on the liver, the most important liver biomarker with liver histology was performed to check the fat deposition in the liver. Furthermore, gene expressions like PPAR-γ, LPL, and FAS were checked.

The present study was aimed to determine the bodyweight reduction effect of *E. cava* 70% ethanol extract, as well as an important obesity-related factor, plasma biomarkers, especially related to hunger and liver biomarkers, inflammatory biomarkers were evaluated in this study to confirm the anti-obesity effect of *E. cava* in HFD-induced obesity.

## 2. Materials and Methods

### 2.1. E. cava and Animals

The *E. cava*, harvested from the coast of Jeju Island of Korea was purchased from a wholesaler. Four weeks old 48 male albino rats of the Sprague–Dawley (SD) strain (70–80 gm) were obtained from Orient Bionics. (Korea; Charles River Technology). The experiment was conducted according to the “animal care and use committee of Kyungpook National University, Daegu, South Korea” (Approval number, KNU2020-11) approved protocol. Furthermore, all in vivo experiments were conducted according to international guidelines for the care and use of laboratory animals [21].

### 2.2. Extraction and Characterization of E. cava

*E. cava* was washed with distilled water and air-dried at 50 ℃. The dried *E. cava* was milled in a grinder (RT-04, Mil Powder Tech Co., Ltd., Tainan Hsien, Taiwan) and passed through a sieve (40 mesh). Dried *E. cava* powder was extracted using water and 70% ethanol (1:20, *w*/*v*) at 100 ℃ for 4 h, and the extract was filtered and then concentrated (N-1N, Eyela Co., Tokyo, Japan) under reduced pressure. Afterward, yield, total phenol, total flavonoid, total sugar, total protein content, and phytochemical composition of extracts were determined.

### 2.3. Gas Chromatography/Mass Spectrometry Analysis (GCMS) of E. cava

*E. cava* GC/MS analysis was performed at Kyungpook National University by the “Center of Scientific Instruments” using an HP 6890 Plus GC gas chromatograph with an HP 5973 mass selective detector (Hewlett-Packard, Palo Alto, CA, USA). The dilution of the sample was prepared by v:v ratio (1:1000) with DCM (HPLC grade). For this, 1 µL of the sample was injected into the HP-5 column. The GC oven temperature was set at 50 ℃ for 4 min, increased to 280 ℃ at a rate of 4 ℃ per minute, and held at the final temperature for 2 min. The carrier gas He (99.99%) velocity was 0.7 mL/min. Quantitative analysis was performed by using the area normalization method.

### 2.4. Cell Culture (In Vitro Study)

The 3T3-L1 pre-adipocyte cells (Korean cell line) were cultured in Dulbecco’s modified Eagle’s medium (DMEM) supplemented with 10% inactivated fetal bovine serum (FBS), L-glutamine (2 mM), streptomycin (10 mg/mL), and penicillin (100 IU/mL). Cells were incubated at 37 ℃ with 5% CO_2_ in a humidified atmosphere. The cells’ media were changed every 2 days. The cells were cultured in 24 well plates for adipocyte differentiation, and when fully confluent after 2 days, they were treated with a differentiation medium containing 0.5 mM IBMX (3-Isobutyl-1-methylxanthine), 1 µM dexamethasone (DEX), and 1 µg of insulin. The medium was changed to DMEM containing 1 µg/mL of insulin after 2 days, and *E. cava*’s final concentration of 500, 100, and 50 µg/mL was added to examine the inhibitory effects on adipocyte differentiation of *E. cava* extract. Fenofibrate was used as a positive control.

### 2.5. Oil Red O Staining

Oil Red O staining was performed as previously described by Mandrekar [22] with a few modifications. Furthermore, 3T3-L1 cells were washed with phosphate buffer saline (PBS) and then fixed in 10% formalin in PBS for 30 min at room temperature. Cells were washed with PBS, and after that, they were stained for 1 h at room temperature in freshly diluted Oil-red O solution.

### 2.6. Cell Viability Determination via MTT Assay

The mitochondrial-dependent reduction of MTT to formozan was determined by a rapid colorimetric assay that measured cell respiration as an indicator of cell viability, following the previously described procedure [23]. For the determination of cell viability, pre-adipocytes 3T3-L1 cells were cultured in the flat bottom 96-well plate at the concentration of 1 × 10^5^ cells/well and treated with *E. cava*. As a result, 50 µL of MTT solution (2 mg/mL) was added to each well of 96-well flat bottom plate and incubated for 4 h at 37 ℃ with 5% CO_2_. The supernatant was carefully aspirated and the insoluble formozan product was dissolved in 200 µL of DMSO (Sigma-Aldrich Chemical, St. Louis, MO, USA). The OD of the culture well was measured using a microplate reader at 540 nm, for absorbance. The final findings were calculated as a percentage cell viability using the following formula: Cell viability % = (OD value of sample/OD value of control) × 100. The experiment was performed in triplicate and results were presented in percentage.

### 2.7. Antioxidant Activity of E. cava (DPPH and ABTS Assay)

The DPPH assay and ABTS assay procedures were followed, as previously reported by Boby [24]. The antioxidant components of *E. cava* were tested for free radical scavenging activity using the DPPH (2,2-diphenyl-1picryl-hydrzyl) and ABTS (2,2-azino-bis(3-ethylbenzothiazolin)-6-sulfonic acid) methods. Different quantities of *E. cava* (1000, 100, 50, 10, 5, and 1 µg/mL) were combined in a 96-well plate with 0.1 mM DPPH solution and ascorbic acid as a positive control in the DPPH test. After incubation, the absorbance was measured using a spectrophotometer at 517 nm (EPOCHTM-2, BioTek Instruments, Seoul, Korea). The data were provided as a percentage, and the final results were used to derive the IC50 value.

In the ABTS radical scavenging activity, 200 µL of freshly prepared ABTS solution was poured to a 96-well plate, followed by 10 µL of *E. cava* extract in each well. The absorbance was measured at 734 nm after 7 min of incubation at room temperature. The final findings were calculated as a percentage of ABTS radical scavenging activity using the following formula: ABTS radical scavenging activity % = [1 − ODcontrol − ODSample/ODcontrol) × 100. The antioxidant potential was measured using the ABTS test activity, which was computed in triplicate measurements.

### 2.8. Experiment Design for In Vivo Study and Animal Housing

The total number of rats (*n* = 48) used in the present study was calculated by the G*power program based on effect size (0.5), α-error probability (0.05), power (1-β error probability) (0.6) numerator df (degree of freedom) (7), and the number of groups were 6. After 7-days of the adaptation period, 48 SD male rats were randomized into six groups (8 rats/group) as Group NC (normal control group), HFD (high-fat diet group), PC (positive control group), T1 (low dose of ECE treatment group), T2 (intermediate dose of ECE treatment group), and T3 (high dose of ECE treatment group). For 7 days prior to the experiment, the animals were acclimatized in a controlled room at 20–25 °C temperature and 55 ± 10% of relative humidity with 12 h light/12 h dark cycles. The animals were allowed to access sterile water and a standard pellet diet *ad libitum*. The animal experimental protocol was approved (Approval number, KNU2020-11) by the animal care and use committee of Kyungpook National University, Daegu, Korea.

### 2.9. Animal Feed and Administration

All groups except the NC group and the HFD group were prepared by adding 125 mg, 250 mg, and 500 mg per kg of feed to 45% fat feed with 3% spray-dried powder mixed with *E. cava* ethanol (70%) extract and polysaccharide. As for the added dose of *E. cava* ethanol (70%) extract and polysaccharide mixed spray-dried powder, rats weigh 100 g and consume 10 g of feed per day. To administer 500 mg/kg/day, it was calculated as 50 mg/10 g feed/100 g rat. In detail, the feed contains a total of 4.6 Kcal/g coming from fats (44.8%), carbohydrates (36.2%), and proteins (19.0%). The experiment was conducted for a total of 8 weeks. Feed intake and body weight were monitored daily. The dose information and daily feed intake for each group are summarized in (Table 1). Test substances were newly prepared immediately before administration, and each administration was performed based on the most recently recorded body weight. The test substance was administered at 10 o’clock, a fixed time every day.

### 2.10. Animal Feeding and FCR

Treatment groups received 125 mg/kg body weight (T1), 250 mg/kg body weight (T2), and 500 mg/kg body weight (T3) of *E. cava* ethanol (70%) extract (ECE) accompanied with HFD (adjusted calories diet (45/FAT) TD-06415). Diet with high-fat (45%) was given to the HFD group, and the NC groups received a fat-free diet. Moreover, 250 mg/kg B.W of Garcinia extract (HCA) was given to the PC group. Rats were treated for a total of 8 weeks. Feed intake and body weight were monitored each day. The test agents were freshly prepared just before administration, and each administration was done based on the most recently recorded body weight. At the end of the experiment, the feed conversion rate (FCR) was calculated from the total consumption of feed divided by the total body weight gain of eight rats for 8 weeks.

### 2.11. Blood Glucose Level Determination

All the rat groups fasted before the blood glucose level determination test. The blood glucose level was determined for each group of rats using the ACCU-Check^®^ Instant blood glucose meter (Roche Diabetes Care GmbH, Altek [Kunshan] Corp. Ltd., Kunshan, China).

### 2.12. Necropsy and Sample Collection

At the end of the experiment, rats were euthanized by carbon dioxide inhalation [25]. The lid was placed over the cage. CO_2_ was delivered from the tank at a flow rate of 10–30% per minute. Finally, animals were monitored for cessation of respiration and left in the chamber for at least 1 min after respiration had ceased. Blood was collected through cardiac puncture and immediately centrifuged at 10,000 RPM for 5 min. Plasma was separated and kept at −70 °C until needed. In addition, pieces of liver and different types of fat tissues, including epididymal, mesenteric, peritoneal, and subcutaneous, based on their sites, have been isolated and preserved in formaldehyde for histopathological analysis, and a small piece of epididymal fat and liver was immediately processed for RNA extraction. Moreover, other major organs, including the kidney and spleen, have also been weighed.

### 2.13. Evaluation of Plasma Level of Various Obesity-Related Biomarkers

The protective effects of the *E. cava* on HFD-induced obesity were determined by measuring the activities of incretins (leptin and ghrelin), insulin, gastric inhibitory peptides (GIP), and gamma-glutamyl transferase (GGT). Additionally, the plasma levels of liver function enzymes (AST, ALP, ALT, and GGT) and lipids, including, low-density lipoprotein (LDL), high-density lipoprotein (HDL), triglycerides (TG), and free fatty acid (FFA), were determined by using enzyme-linked immunosorbent assay (ELISA) kits (Sigma-Aldrich, St. Louis, MO, USA and Abcam, Cambridge, UK) following the manufacturer’s instructions. Furthermore, the plasma level of TNF-α and IL-6 were determined.

### 2.14. Liver and Adipose Histopathology

Liver histological analysis has been performed to determine the effects of *E. cava* on hepatocytes. For this purpose, 10% of buffered formalin was used for embedding the small portion of liver tissues. A small portion of mesenteric and epididymal adipose tissues was collected from all rats after euthanasia. Adipose tissues were fixed in 10% buffered formaldehyde and then processed with hematoxylin and eosin to assess the morphology.

### 2.15. Extraction of Total RNA and PCR Analysis

The effects of *E. cava* on the expression of genes that are essential to regulate the activities of enzymes involved in adipocytes and lipid metabolism were determined using quantitative real-time PCR. For this purpose, liver tissue was homogenized (IKA T10 basic Homogenizer, Seoul, Korea), and total RNA was extracted according to the protocol for the TRIzol reagent. The extracted RNA was diluted 2-fold using DEPC-treated water. The concentration (µg/mL) and purity of RNA were determined using a U-2800 spectrophotometer (Hitachi High Technologies, Tokyo, Japan). Complementary DNA was synthesized from 100 ng of RNA by using Superscript^®^ III First-Strand Synthesis SuperMix (Life Technologies, Carlsbad, CA, USA), according to the protocol. The primers used to detect the target gene expression and their corresponding annealing temperatures are listed in (Appendix A).

### 2.16. Receiver Operating Characteristics (ROC) and Area under ROC Curve (AUROC)

ROC is considered the gold standard in diagnostic observations. The body weight data were analyzed through the ROC curve. An observation about treatment and non-treated groups was analyzed with an adiposity index. The adiposity index was calculated as previously described by [26], and we found AUROC to check the accuracy of our observation in this experiment. The area under the ROC curve has a useful interpretation for disease categorization from healthy subjects. A ROC graph was plotted between sensitivity vs. 1-specificity of diagnostic observation [22,27].

### 2.17. Statistical Analysis

All the values in the results were expressed as mean ± SEM (standard error of the mean). Statistical analysis was done by one-way analysis of variance (ANOVA) using the SAS program and Graph Pad Prism statistical software, Version 5.01 (Graph Pad Software Inc., La Jolla, CA, USA) with the comparison between groups. *p*-values less than 0.05 were considered significant.

## 3. Results

### 3.1. Extraction and Characterization of E. cava

The method used for the preparation of *E. cava* ethanol (70%) extract had a yield of 23.47% with total phenol, total flavonoid, total sugar, and total protein content of 45.61 ± 1.01 TA g/100 g, 21.20 ± 0.99 rutin g/100 g, 13.19 ± 0.51 glucose g/100 g, and 14.90 ± 0.54 BSA g/100 g, respectively. Phloroglucinol and dieckol were the phlorotannins detected extract and their content ranged from 0.15 to 0.78, and 18.75 to 24.39 mg/g, respectively.

### 3.2. Gas Chromatography/Mass Spectrometry Analysis (GCMS) of E. cava

The GCMS analysis of *E. cava* revealed that bioactive compounds present in the extracts have a broad range of biological activities. The GCMS chromatographic analysis of the 70% ethanolic extract of *E. cava* illustrated in Figure 1 shows 11 peaks, indicating the presence of different phytochemical components. Of the 11 different compounds identified in the GCMS analysis, we found nine compounds with multiple biological activities. The compounds have strong antioxidant, anti-inflammatory, and antibacterial activities. Some of the compounds have anti-obesity activities or are abundantly present in plant extracts, which have been previously reported for their anti-obesity effects. The details of compounds and their activities are given in (Appendix A).

### 3.3. Suppressive and Cytotoxic Effects of E. cava Extract on Adipogenesis in 3T3-L1 Pre-Adipocytes

3T3-L1 differentiated adipocytes were treated with fenofibrate (FF) positive control and *E. cava* extract, in the presence of insulin 1 µg/mL, and accumulated lipids droplets were visualized by Oil Red O staining under the light microscope (Figure 2A). The *E. cava* extract suppressed the differentiation and decreased the accumulation of lipid droplets in dose-dependent manners (Figure 2B). The cytotoxic effects of *E. cava* on the 3T3-L1 cells were determined using the MTT cell viability assay. The *E. cava* did not seem to have any cytotoxic effect on cells (Figure 2C).

### 3.4. Antioxidant Activity of E. cava (DPPH and ABTS Assay)

*E. cava* exhibited the DPPH and ABTS radical scavenging activity in concentration-dependent manners from 1000 to 0.1 µg/mL (Figure 3). *E. cava* has a significant IC50, indicating that it is a strong antioxidant. *E. cava* had an IC50 of 75.48 µg/mL in the DPPH assay and 70.44 µg/mL in the ABTS assay, which is good in both assays. Our findings indicate that *E. cava* is an effective antioxidant.

### 3.5. Observation of General Body Weight and Feed Conversion Ratio (FCR)

To examine the effect of *E. cava* extract with HFD on body weight, we measured the weight in all groups of rats. There was an increase in body weight with time among all groups, but the trend was different between different groups (Figure 4A). Feed intake varied between different groups (Figure 4B). As a result, when comparing the body weight change of the NC and the HFD group, the NC had 23.77% less body weight in comparison with the HFD group. However, the body weight decreases in a dose-dependent manner. The highest dose treatment group of *E. cava* with HFD (500 mg) was 16.72% less body weight as compared to the HFD group. The bodyweights of the treatment 1 and treatment 2 groups were 12.1% and 16.64% less than those of the HFD group, respectively. On the other hand, the FCR for the NC and *E. cava*-HFD fed rats were 32.3 and 12.34% higher than that of HFD-fed rats, respectively. The normal diet group FCR was higher compared to other groups (Table 2). This shows that normal feed consumption has no negative effects on animal health, but the continuously high calories feed consumption will lead to obesity and obesity-related complications.

### 3.6. Effects of E. cava Treatment on Organ Weight

The overall increase in body weight will affect the visceral organs [28]. The increase in body fat will lead to fatty liver disease. After 8 weeks of *E. cava* treatment, we observed a significant reduction in body weight. Furthermore, the organ weights of all groups were measured to figure out the effects of *E. cava* treatment. As demonstrated in Figure 4C–E (Appendix A), there was a clear distinction in the weight of the liver, spleen, and kidney among treatment and non-treatment rats. The average liver weight of each rat in the HFD group was 17.82 g, and the average liver weight of NC and T3 liver weights were 12.67 and 12.76 g, showing a significant effect of *E. cava* on the reduction of live fat deposition. The positive control group’s liver weight was 13.89 g, which was similar to the T2 group’s weight of 13.84 g. The T1 group’s liver weight was 14.78 g, which was 17% less than the HFD group. The weight of the spleen was reduced (18.5%) by *E. cava* treatment compared with the HFD group. Kidneys play a key role in several body functions, and obesity increases the risk of kidney-related issues [29]. The HFD increases the weight of the kidneys as compared to normal control, but the upper limit of the maximum reduction of kidney weight with maximum dose was 14% less than the HFD group and was comparable with the NC group. The difference in kidney weight between the NC and T3 groups was that the T3 group was just 2.5% higher, which was closer to the normal value.

### 3.7. Effects E. cava Treatment on Visceral Fat

Deposition of fat in the visceral area can cause serious health issues such as cardiovascular disease, hypertension, and type 2 diabetes. While checking different parameters, it is necessary to check the effect of an anti-obesity drug on visceral fat. The adipose tissues weight, including subcutaneous, epididymal, perirenal, and mesenteric, were high in HFD-fed rats as compared to the ND group. The HFD group subcutaneous, mesenteric, peritoneal, and epididymal fat weights were 14.9, 8.4, 9.14, and 7.46 g and the ND group weights were 4.25, 3.75, 4.13, and 3.42 g, respectively. The ND group subcutaneous, mesenteric, peritoneal, and epididymal fats were 67.11, 55, 54, and 54.15%, respectively, lesser than the HFD group. The adipose tissues weights were decreased in EC-supplemented HFD-fed rats compared to rats fed only HFD as demonstrated in (Figure 5; Appendix A). The *E. cava* treatment group T3 subcutaneous, mesenteric, peritoneal, and epididymal fats were 56.44, 45, 36.10, and 36.32%, respectively, less than HFD. These results indicate that the *E. cava* extract successfully decreases adipogenesis in the body.

### 3.8. Effect of E. cava on Adipokines, Incretins, Total GIP, and Insulin

The obesity-related important plasma biomarkers were analyzed through ELISA. Obesity due to high-energy diet intake causes hormonal imbalance in the body. Incretins level in the body are important to determine obesity. Both leptin and ghrelin are related to dietary intake and appetite control. Long-term regulation of body weight and dietary intake while ghrelin plays an important role in hunger and meal initiation. It is also predicted that ghrelin is associated with insulin resistance [30]. It is reported by Gil-Campos et al. and Zou et al. [30,31] that the increased level of leptin and low level of ghrelin are associated with obesity and hyperinsulinemia.

In our results, in HFD diet rats without treatment, the leptin level was high, and they had a low level of ghrelin, as well as the levels of insulin and glucose were higher as compared to other groups. There is no significant difference between the results of positive control, normal control, and EC-supplemented HFD (500 mg/kg). The level of leptin was significantly reduced by treatment groups as dose-dependent manners and the level of ghrelin was high with a clear decrease in insulin and blood glucose level. The level of GIP is also correlated with insulin. The GIP level was reduced by treatment groups and high in the HFD group. The impact of EC on total GIP, leptin, ghrelin, and was insulin in rat plasma is given in Figure 6.

### 3.9. Effects of E. cava on Blood Glucose Level

The glucose level in the blood and insulin correlate with each other. The raised level of insulin may indicate insulin resistance or be a sign of type 2 diabetes. For insulin resistance-related effects of EC water extract, we have measured blood glucose levels. The NC group blood glucose level was 102.3 mg/dL, and the HFD blood glucose level was 139.6 mg/dL, which was 36.4% higher than the normal group, showing the prediabetes condition of rats. According to research [32], the normal blood glucose upper limit range for rats is 6.2 mmol/L or 111.6 mg/dL, which may vary in different breeds, but it is also comparable with humans. Here, the results show (Table 2) that the treatment group’s blood glucose was at a normal level. The high dose group (T3) showed a blood glucose level of 102 mg/dL, which was the same as the normal diet group. The low dose (T1) and medium-dose (T2) blood glucose levels were 105.4 mg/dL and 104.4 mg/dL, respectively. Blood plasma glucose levels were significantly decreased in EC-fed rats compared with HFD-fed rats at 4 weeks of treatment.

### 3.10. Effect of E. cava on Plasma Level of Pro-Inflammatory Cytokines

According to different studies, obesity is related to low-grade chronic inflammation and is characterized by abnormal cytokine production [33]. In obesity conditions, adiposities release pro-inflammatory cytokines such as TNF-α and IL-6 [34]. In vitro studies show the anti-inflammatory activity of *E. cava* [35]. Feeding of HFD increased the TNF-α and IL-6 but was significantly reduced by the supplementation of EC (125–500 mg/kg) in HFD. The impact of EC on TNF-α and IL-6 of rat plasma is shown in Figure 7. Decreased cytokines production is another strong evidence indicating that the *E. cava* treatment is reducing the adipose tissue size by regulating fat deposition as well as decreasing the inflammatory condition in the body due to obesity.

### 3.11. Effects of E. cava on Lipid Profile and Liver Function Biomarkers

The increase in body weight is the major reason for the change in the lipid profile. In particular, an increase in LDL, FFA, and TG and a decrease in HDL level are serious outcomes of obesity. Furthermore, increasing the level of total cholesterol will increase the risk of coronary heart disease [36]. The level of TG directly correlates with obesity. The impact of *E. cava* on the plasma lipid profile is presented in Figure 8 (Appendix A). Elevated concentrations of TC and LDL indicated hyperlipidemia in HFD-fed rats. On the other hand, treatment of *E. cava* (125–500 mg/kg b.w) to rats fed with HFD significantly reduced the plasma levels of these parameters and the level of lipids is the same as that of normal rats. Similarly, *E. cava* treatment significantly lowered the atherogenic index (AI) in HFD-fed obese rats (Figure 8F) (Appendix A). The atherogenic index is strongly correlated with cardiovascular disease (CVD) risks, which is the log ratio of plasma concentrations of TG and HDL [37].

Furthermore, liver enzymes alkaline phosphatase (ALP), alanine transaminase (ALT), aspartate transaminase (AST), and gamma-glutamyl transferase (GGT) were used to estimate live functions. Previously, a several pieces of research were conducted to assess the link between ALT and GGT with obesity. Researchers have established a link between abdominal obesity and ALT and GGT [38,39,40,41]. The ALT level is considered a specific marker for hepatic dysfunction and is found primarily in this organ, whereas GGT has significant activity in the kidney and liver [42,43]. In our study, the liver enzymes showed a strong relation with obesity. The results are shown in Figure 9 (Appendix A). The level of liver enzymes in the HFD group without treatment was elevated, but the AST, ALT, ALP, and GGT levels were significantly reduced by *E. cava* treatment.

### 3.12. Effects of E. cava on Hepatocytes and Adipocytes

The histopathological changes in the liver tissues of rats linked with the feeding of HFD and HFD accompanied by treatments (HCA and EC) are summarized in Figure 10. In the HFD group, the accumulation of lipid droplets was increased, whereas, in the treatment with HCA-HFD and EC-HFD group rats, it was decreased. The 8 weeks feeding of HFD generated the accumulation of fat droplets in the hepatic tissues of rats. A substantial increase in the accumulation of fat droplets was observed in the livers of the HFD-fed rats compared to the ND-fed rats. Rats fed a control diet reveal the normal structure of the liver. The effects of HCA (positive control) and *E. cava* on mesenteric adipose tissues and the size of epididymal adipocytes are also obvious from Figure 10. The adipocytes’ size was markedly increased in HFD-fed rats as compared to the ND group. In contrast, *E. cava*-supplemented HFD reduced the size of the adipocytes in the previously mentioned fat tissues of rats.

### 3.13. Effects of E. cava Extract on Adipogenic and Lipogenic Gene Expression

Total mRNA was extracted from different groups of rats, and qualitative gene expression was assessed by polymerase chain reaction (or endpoint PCR). *E. cava* induced reduction in lipid accumulation, which was measured using the mRNA markers related to lipogenesis and adipogenesis. Results obtained from different groups show that the high-fat diet induced the up-regulation of fatty acid synthase (FAS), lipoprotein lipase (LPL), SREBP-1C, and peroxisome proliferator-activated receptors Gama (PPAR-γ) genes (Figure 11), *E. cava* supplementation T3 the expression of the key transcription factors PPAR-γ was same as in NC and PC groups. The expression of FAS, LPL, and SREBP-1c also decreased in a dose-dependent manners.

### 3.14. Receiver Operating Characteristics (ROC) and Area under the ROC Curve

In the experiment, we had a normal diet group, a high-fat diet group, a positive control treatment with HFD, and three different groups treated with *E. cava* (125 mg, 250 mg, and 500 mg/kg b.w), and HFD. To confirm our diagnostic observations related to non-obese and obese rats, we analyzed our data with receiver operating characteristics (Figure 12). The adiposity index was calculated using the formula (Body fat/body weight × 100) [26], and adiposity index was analyzed using ROC the results shown in the (Figure 12b). The AUROC was calculated through the trapezoidal method and the AUROC = 0.92, which shows 90% accuracy in our results.

## 4. Discussion

In the present study, we examined the anti-obesity efficacy of an extract from *E. cava* 70% ethanol extract. *E. cava* is enriched with different bioactive compounds, which have a wide range of biological activities. In the GCMS analysis of *E. cava* extract, different compounds were found. The major activities of these compounds, as reported recently, are antibacterial, antioxidant, anti-inflammatory, anti-obesity, antifungal, and anticancer. Among these compounds, some of them have directly reported activities, but some of them are abundantly present in different types of extracts that have significant biological activities. Some chemistry studies show that the combinations of these compounds with other compounds and the derivatives of these compounds have different biological and nonbiological activities. We can conclude that the *E. cava* profile shows it has strong anti-inflammatory, anti-obesity, and antioxidant properties.

The in vitro 3T3-L1 pre-adipocytes, the in vitro results show that the adipocytes’ differentiation increases with a decrease in concentration, and the cytotoxic effect of the extract is very small.

The in vitro antioxidant activity of *E. cava* shows strong antioxidant effects. It was important to figure out the antioxidant activity in the obesity-related study because of the strong relationship between obesity and oxidative stress. Oxidative stress has long been known to have a role in the pathogenesis of a variety of diseases, including cardiovascular disease, diabetes, and obesity. Obesity-induced oxidative stress has been postulated as the source of oxidative stress in adipose tissue, which is then transported to other tissues and may account for obesity-related disorders [44]. Our results show that *E. cava* has good antioxidant properties and decreases obesity. It will help to reduce oxidative stress, which is strongly related to obesity. We can use *E. cava* as an antioxidant to help lower oxidative stress. *E. cava* antioxidant activity helps to minimize oxidative stress, thus, we may use it for that purpose.

In an in vivo study of 8 weeks, Sprague–Dawley rats were treated with *E. cava* doses of 125, 250, and 500 mg/kg/day with HFD. It is proven that HFD causes obesity and metabolic diseases in rodents [45]. After continuous treatment for 8 weeks, we observed a significant anti-obesity impact of *E. cava* in HFD-induced obese rats (Appendix A). When compared to HFD-fed rats that did not receive *E. cava* supplementation, rats receiving *E. cava* administration showed substantial reductions in body weight as well as live, subcutaneous, peritoneal, epididymal, and mesenteric fat weights.

The increase in overall body weight effects on visceral organs negatively, especially in the liver due to the accumulation of fat [28]. In particular, the HFD group’s liver weight was higher as compared to the *E. cava*-supplemented HFD treatment groups. Bodyweight loss was accompanied by a decrease in adipose tissue weight. Our findings also show that the weight of adipose tissues in rats treated with EC-supplemented HFD was much lower than in the HFD treatment groups. Furthermore, from the body weight and adipose tissue data, the body index was calculated and analyzed with ROC and AUROC. The AUROC value was 0.902, which indicates 90% accuracy.

In this study, we also investigated the changes in plasma levels of obesity-associated factors, including leptin, ghrelin, glucose, insulin, gastric inhibitory polypeptide, and pro-inflammatory cytokines levels, as well as the lipid profile, including HDL, LDL, FFA, TG, and TC, when the atherogenic index was calculated. In addition, the plasma concentrations of liver biomarkers, especially those related to obesity, such as AST, ALT, ALP, and GGT, were examined through ELISA.

Excessive body fat disturbs organ functions as well as body hormones. Leptin and ghrelin hormones are directly related to suppressing and increasing the desire for food intake, and obesity is also known to be related to leptin resistance. Abnormalities in these kinds of hormones should be one of the major causes of obesity [46]. Results show that in the HFD-obese group, the leptin level was higher, which led to leptin resistance. However, the EC-supplemented groups showed the normal level of leptin. In the case of ghrelin, the level was lower in the HFD obese group, but this effect was also reversed by *E. cava* treatment.

Obesity and blood glucose are linked negatively to each other. Multiple metabolic alterations accrue in the body due to obesity, and these are the risk factors for glucose homeostasis abnormalities [47,48]. Insulin is an important hormone that helps to maintain the glucose level in the body. Different studies show that obesity and type 2 diabetes are directly linked with insulin resistance [49]. When we compared the blood glucose levels of HFD groups with EC-supplemented HFD groups, we found that the glucose level of the HFD group was markedly higher than the EC-supplemented group. The level of insulin was high in the HFD obese groups, which indicates that the high level of glucose present in the body is due to insulin resistance in rats. Our findings show that the EC extract has the ability to control insulin resistance, which is the major cause of type 2 diabetes. Here, in EC-supplemented groups, HFD groups show a normal level of insulin, which shows that the extract helps to regulate the insulin release mechanism normally. Gastric inhibitory polypeptide (GIP) hormone is released in the response of feed intake, which also causes insulin release. Evidence suggests that a key link between obesity development, type 2 diabetes, insulin resistance, and the consumption of energy-rich high-fat diets is the GIP receptor-mediated effects [50]. The GIP level was significantly reduced in EC-supplemented HFD groups.

Previously, adipose tissues were thought to be only long-term energy storage sites, but concepts have changed, and new studies indicate that they plan a key role in systemic metabolism integration. It has been proven that HFD can develop a pro-inflammatory state. Obesity conditionally upregulates the production of most adipokines. These pro-inflammatory proteins typically function to promote obesity-linked metabolic diseases. Recently, some TNF and IL-6 adipokines have been identified that promote inflammation. The level of TNF in adipose tissues and plasma is associated with body weight. Levels of TNF in the blood were found to positively correlate with markers of inflammation. In the case of IL-6, the plasma level of IL-6 increased in type 2 diabetic patients. It is estimated that approximately one-third of the total circulating IL-6 is produced by adipose tissues [51]. According to our findings, HFD caused an increase in body weight, which also caused an increase in plasma TNFα and IL-6 levels, but the level of TNF-α and IL-6 level is significantly reduced in EC-supplemented HFD-fed rats groups. This is a good indicator, showing us that the extract is positively controlling the pro-inflammatory changes in the body.

The liver is the major organ of the body that is responsible for the detoxification and production of proteins, and a bundle of enzymes are involved in this process. The levels of enzymes and proteins in the blood that are produced by the liver can be used to analyze liver health. Similarly, elevated levels of GGT, ALT, AST, and ALP indicate liver injury [52]. Another study shows the increased level of liver enzymes, especially ALP and GGT, is an indicator of nonalcoholic fatty liver disease and liver dysfunction and also relates to type 2 diabetes [53]. In our study, we found elevated levels of AST, ALP, ALT, GGT, and TG in HFD-obese rats. On the other hand, EC-supplemented HFD rats’ enzymes were at a normal level.

Different studies have reported that blood lipids and obesity are closely related [54]. Lipids are an important compound involved in different cellular functions and homeostasis. Obesity disturbs the lipid levels in the body, and these abnormalities show hepatic and cardiovascular diseases. The liver plays an important role in lipid metabolism, synthesis, and transportation. An abnormal lipid profile is the key indicator of liver dysfunction. Likewise, increased LDL level and decreased HDL level are important risk factors of coronary heart disease [55,56]. In our study, HDL levels are high in EC-supplemented HFD rats as compared to HFD obese rats. LDL level is higher in HFD obese rats, and EC-supplemented HFD rats have a normal level of LDL as well as significantly reduce the level of TC and free fatty acid level in plasma. There is also a decrease in the atherogenic index (AI) in EC-supplemented HFD rats compared to the HFD rats’ groups, which also shows that obesity increases the AI level in plasma. AI level increase in plasma is associated with cardiovascular diseases [57]. The decrease in the atherogenic index suggests that *E. cava* ethanol extract may reduce the risk of cardiovascular atherosclerosis.

Histopathological and metabolic changes accrue due to obesity in the liver, such as nonalcoholic fatty liver disease. Under normal conditions, small lipid droplets can be observed only with an electron microscope or fat stains, but under metabolic imbalance conditions, the droplets become larger and can easily be observed under a light microscope. Any change in lipid transport and lipoprotein secretion can cause the enlargement of normal lipid droplets in the liver [58]. The histological study of the liver shows that the lipid droplets were increased in HFD rats but reduced in the EC-supplemented HFD rats’ groups. Overweighting due to a high-calorie diet causes structural and cellular composition changes in adipose tissues. Nonetheless, these changes cause an enlargement of adipocyte size (hypertrophy) [59]. The histopathological study of adipose tissues shows that the HFD groups’ epididymal and mesenteric adipose tissues adipocyte enlarged as compared to the ND group. EC-supplemented HFD rats’ groups showed a tremendous decrease in the size of adipocytes. Thus, our findings show that the EC extract is effective in controlling overall body fat as well as protecting the visceral organs from hypolipidemia through different metabolic pathways regulation.

Adipose tissues play a key role in the body’s energy homeostasis, and thus, the functional disorder can cause serious issues such as metabolic syndromes and diabetes. The transcription factor-like PPAR-γ has been known to regulate adipocyte differentiation, fat storage, and glucose metabolism, and it targets hypolipidemic drugs. In obesity-related studies, the physiological and molecular functions of PPAR co-activators and co-repressors in relation to adipocyte energy metabolism were studied [60]. Lipogenic genes (FAS and PPAR-γ) are regulated by SREBP1 and LPL gene, which controls the lipid metabolism of adipocytes induced by the PPAR-γ activation. Moreover, overexpression of LPL can cause insulin resistance and obesity [61,62]. In our study, the mRNA expression of adipogenic genes, such as PPAR-γ and LPL, was upregulated by the HFD, but the EC water extract supplementation reversed this effect. Furthermore, the mRNA expression of the lipogenic enzyme, FAS, was also downregulated in the EC-supplemented HFD group. Our results suggested that EC can inhibit adipogenesis, followed by TG synthesis.

Thus, together, all these findings suggest that *E. cava* reduces body weight as well as decreases the risk of obesity-related problems by regulating the hormone mechanism and lowering the lipids in plasma (Figure 13), and moreover, by decreasing the expression of adipogenic and lipogenic genes.

## 5. Conclusions

In conclusion, our findings show that *Ecklonia cava* extract inhibits 3T3-L1 adipocyte differentiation and protects rats against high-fat diet-induced obesity. It has strong antioxidant properties. *E. cava* extract can reduce 16.72% of the body weight, as well as a decrease of 28% liver weight compared with HFD. The *E. cava* treatment reduces subcutaneous, mesenteric, peritoneal, and epididymal fats by 56.44, 45, 36.10, and 36.32%, respectively, compared with HFD rats. It regulates the blood glucose level (27% less then HFD) as well as plasma insulin level. It reducess the plasma biomarkers related to obesity (HDL, LDL, FFA, and TG) and lipid accumulation. It also regulates liver biomarkers (ALT, AST, GGT, and ALP), incretins-related effects, and inflammatory biomarker (TNF-α and IL-6) inhibition. It also affects the genes (FAS, LPL, and SRBP-1c) that are directly associated with obesity. The histopathological imaging shows the normal size of hepatocytes and fat cells. In light of our results, we suggested that *E. cava* could be a potential candidate for the prevention of obesity-induced by a high-fat diet.

## Figures and Tables

**Figure 1 antioxidants-11-00310-f001:**
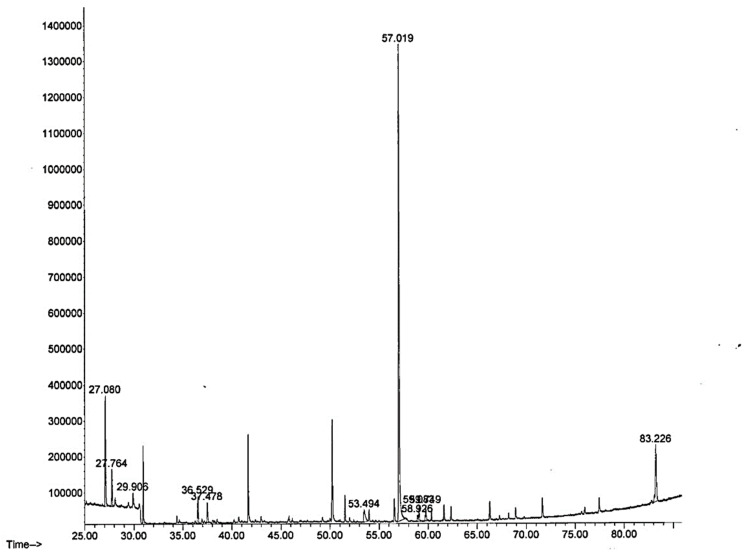
Analysis by gas chromatography and mass spectrometry (GC/MS) of an ethanol extract of *E. cava* (70%). Chromatograms were obtained by GC/MS analysis of *E. cava*. Each peak in the chromatogram represents a specific compound obtained with a unique (RT) retention time.

**Figure 2 antioxidants-11-00310-f002:**
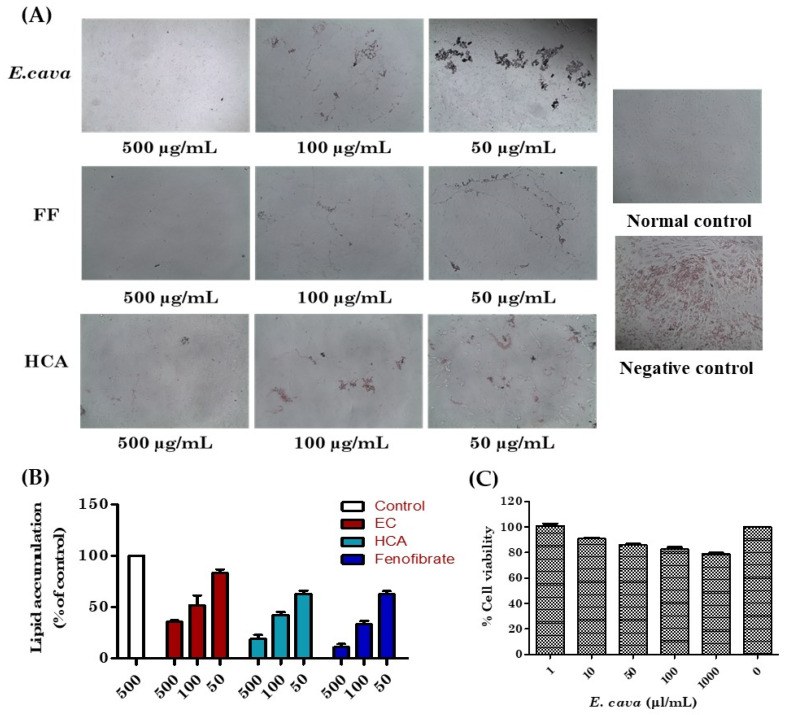
The effects of *E. cava* ethanol (70%) extract on the differentiation and adipogenesis of 3T3-L1 cells. (**A**) Intracellular lipids were stained with Oil Red O, (**B**) lipid accumulation percentage, and (**C**) cytotoxicity of *E. cava* extract in 3T3-L1 adipocytes.

**Figure 3 antioxidants-11-00310-f003:**
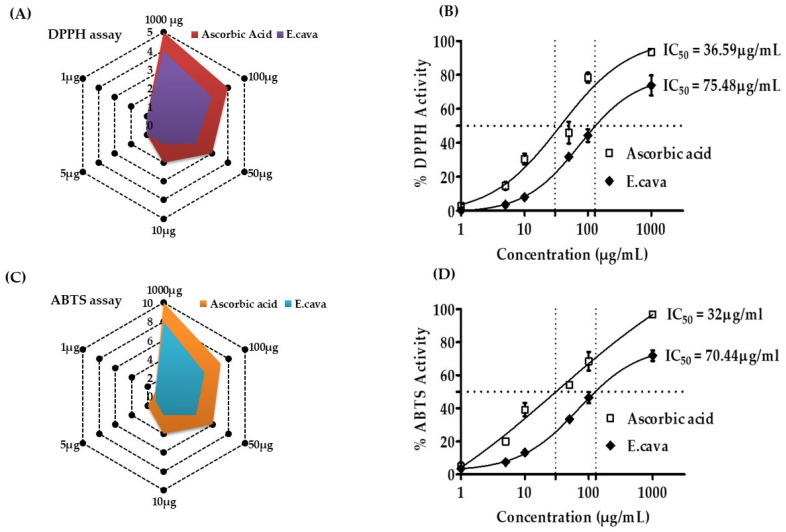
*Ecklonia cava* 70% ethanol extract antioxidant activity in DPPH (2,2-diphenyl-1-picrylhydrayl) and ABTS (2,2-azino-di-(3-ethylbezothiazoline)-6-sulfonic acid assays. (**A**) the % free radical scavenging activity of *E. cava* and (**B**) IC_50_ DPPH assay. (**C**) ABTS scavenging activity with IC 50 (**D**). The absorbance of the sample and DPPH solutions was measured at 517 nm after incubation at 37 °C. Positive control was employed, which was ascorbic acid. After combining the ABTS solution with the samples, the absorbance at 734 nm was measured for the ABTS test. The results are expressed as the mean % inhibition ± SEM and IC_50_ (*n* = 3).

**Figure 4 antioxidants-11-00310-f004:**
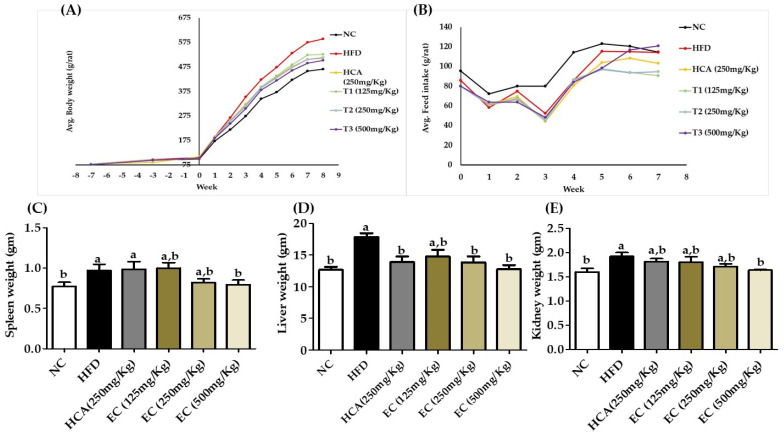
The effects of *Ecklonia cava* ethanol (70%) extract (ECE) on body weight (**A**), average feed intake (**B**), spleen (**C**), liver (**D**), and kidney (**E**) weights in high-fat diet-fed SD rats for 8 weeks. Data are presented as means ± SEM (*n* = 8 for each group). Values with different letters present significant differences (*p* < 0.05).

**Figure 5 antioxidants-11-00310-f005:**
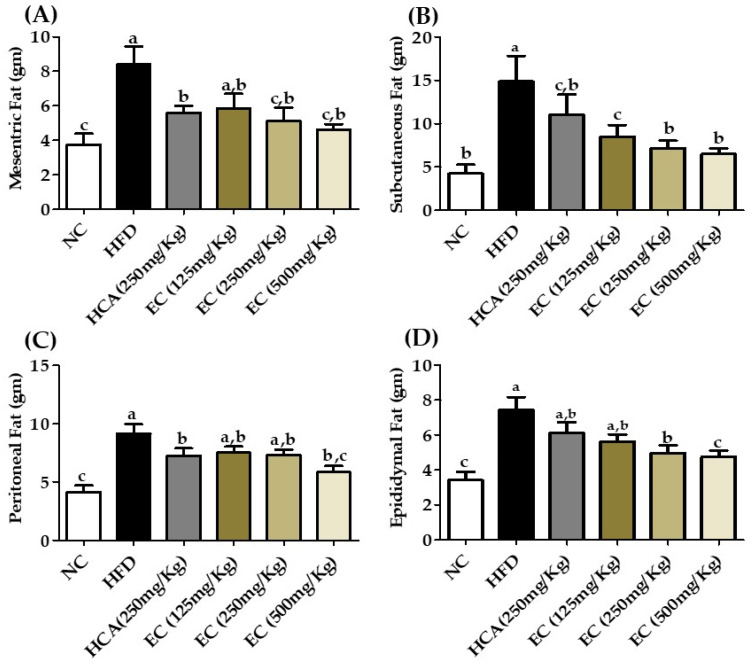
Changes in organ fat, including mesenteric fat (**A**), subcutaneous fat (**B**), peritoneal fat (**C**), and epididymal fat (**D**) weights among rats fed with a normal diet (NC), a high-fat diet (HFD), or HFD accompanied by treatment with ECE at different doses of 125 mg (T1); 250 mg (T2), and 500 mg (T3) per kg BW. Data are presented as means ± SEM (*n* = 8 for each group). Values with different letters present significant differences (*p* < 0.05).

**Figure 6 antioxidants-11-00310-f006:**
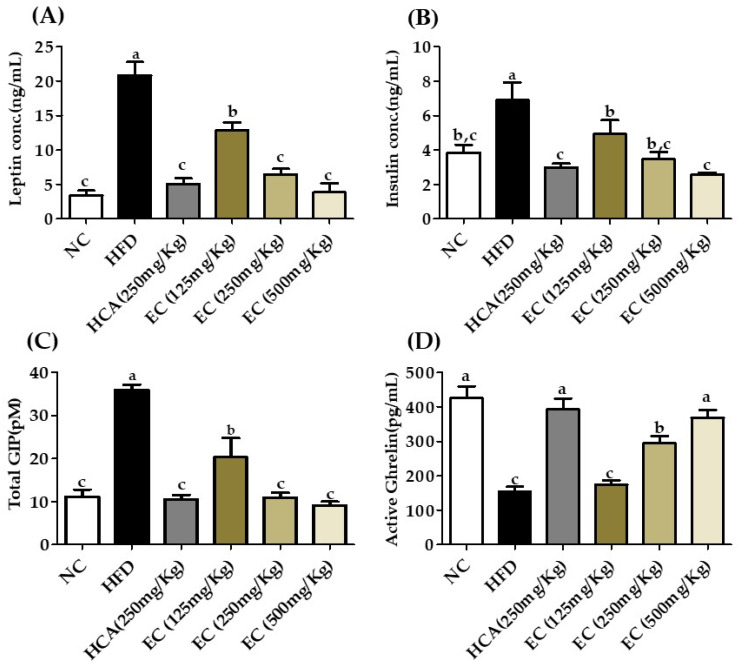
The effects of *Ecklonia cava* ethanol (70%) extract (ECE) on incretins, adipokines, endocrine peptides, and liver function enzymes after 8 weeks of feeding ND; HFD; and 125, 250, and 500 mg/kg b.w of ECE accompanied by a high-fat diet (HFD). (**A**) Leptin, (**B**) active ghrelin, (**C**) total GIP, and (**D**) insulin. Data are presented as means ± SEM (*n* = 8 for each group). Values with different letters present significant differences (*p* < 0.05).

**Figure 7 antioxidants-11-00310-f007:**
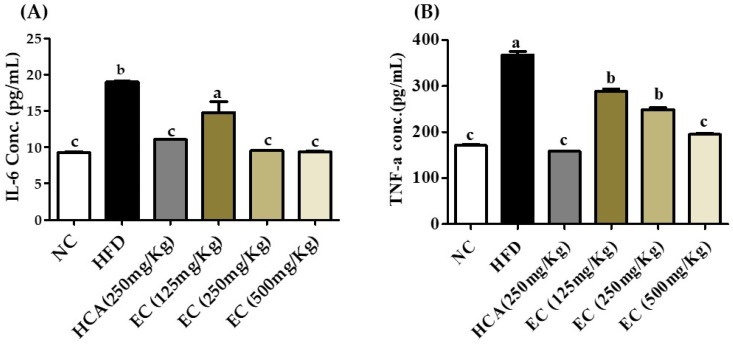
The effects of *Ecklonia cava* ethanol (70%) extract (ECE) on pro-inflammatory cytokines in rat plasma after 8 weeks of feeding ND; HFD; and 125, 250, and 500 mg/kg b.w of ECE accompanied by a high-fat diet (HFD). (**A**) IL-6 and (**B**) TNF α. Data are presented as means ± SEM (*n* = 8 for each group). Values with different letters present significant differences (*p* < 0.05).

**Figure 8 antioxidants-11-00310-f008:**
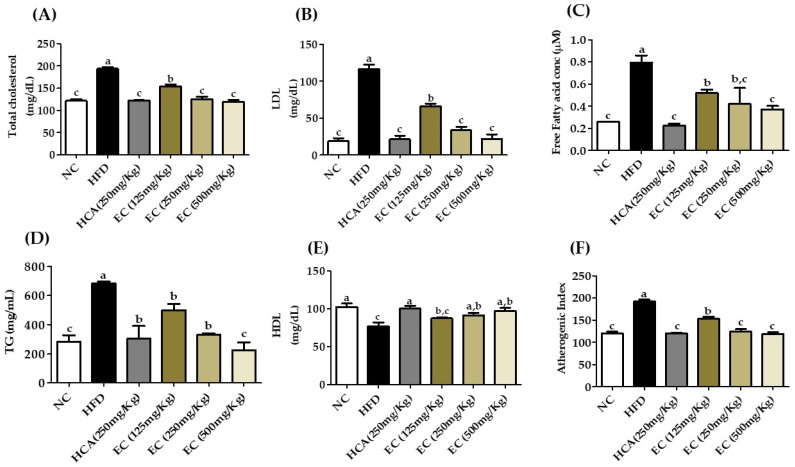
The effects on the lipid profile of rats fed a normal diet and a high-fat diet with or without treatment of different doses of ECE (125, 250, and 500 mg/kg b.w) for 8 weeks. (**A**) Total cholesterol, (**B**) LDL ((low-density lipoproteins), (**C**) free fatty acid, (**D**) triglyceride (TG), (**E**) HDL (high-density lipoproteins), and (**F**) atherogenic index (AI). Data are presented as means ± SEM (*n* = 8 for each group). Values with different letters present significant differences (*p* < 0.05).

**Figure 9 antioxidants-11-00310-f009:**
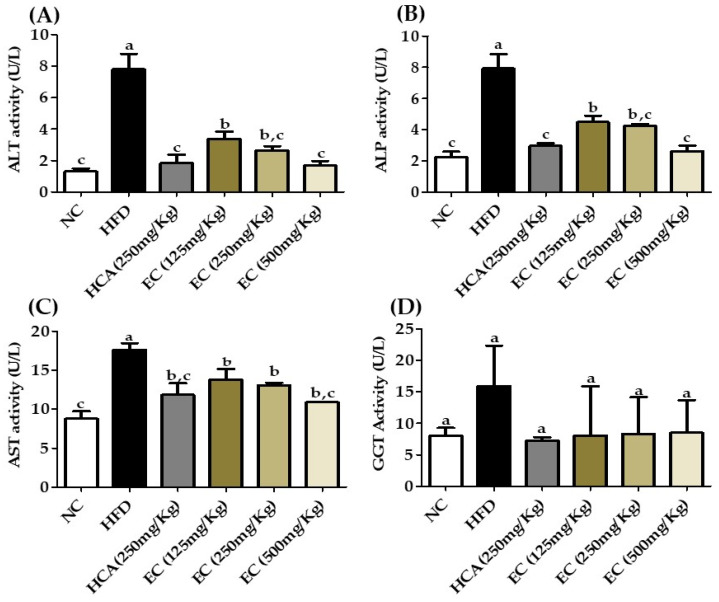
The effects of *Ecklonia cava* ethanol (70%) extract (ECE) on plasma biomarkers as liver function enzymes after 8 weeks of feeding of ND; HFD; and 125, 250, and 500 mg/kg b.w of ECE accompanied by a high-fat diet (HFD). (**A**) alanine aminotransferase (ALT), (**B**) alkaline phosphatase activity (ALP), (**C**) aspartate aminotransferase activity (AST), and (**D**) gamma-glutamyl transferase activity (GGT). Data are presented as means ± SEM (*n* = 8 for each group). Values with different letters present significant differences (*p* < 0.05).

**Figure 10 antioxidants-11-00310-f010:**
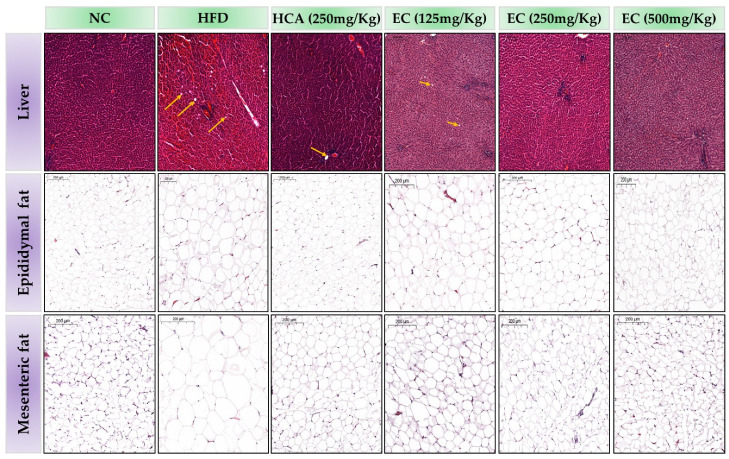
The effects of *Ecklonia cava* (EC) extract on adipocyte accumulation in hepatic tissue, epididymal, and mesenteric adipose tissues (hematoxylin and eosin staining, 10× magnification). The figure shows hepatic tissue, epididymal and mesenteric adipose tissues (from top to bottom) from different groups, including normal diet group (NC), a high-fat diet group (HFD), or groups fed with EC-HFD at different doses of 125 mg (T1), 250 mg (T2), and 500 mg (T3) per kg B.W (moving from left to right). An image of the hepatic tissue of a rat from the HFD group shows fewer/absence of lipid droplets (yellow arrow) in treated groups as compared to the high-fat diet group. Moreover, adipose tissue histology shows a small-sized large number of adipocytes in treated groups as compared to the high-fat diet group.

**Figure 11 antioxidants-11-00310-f011:**
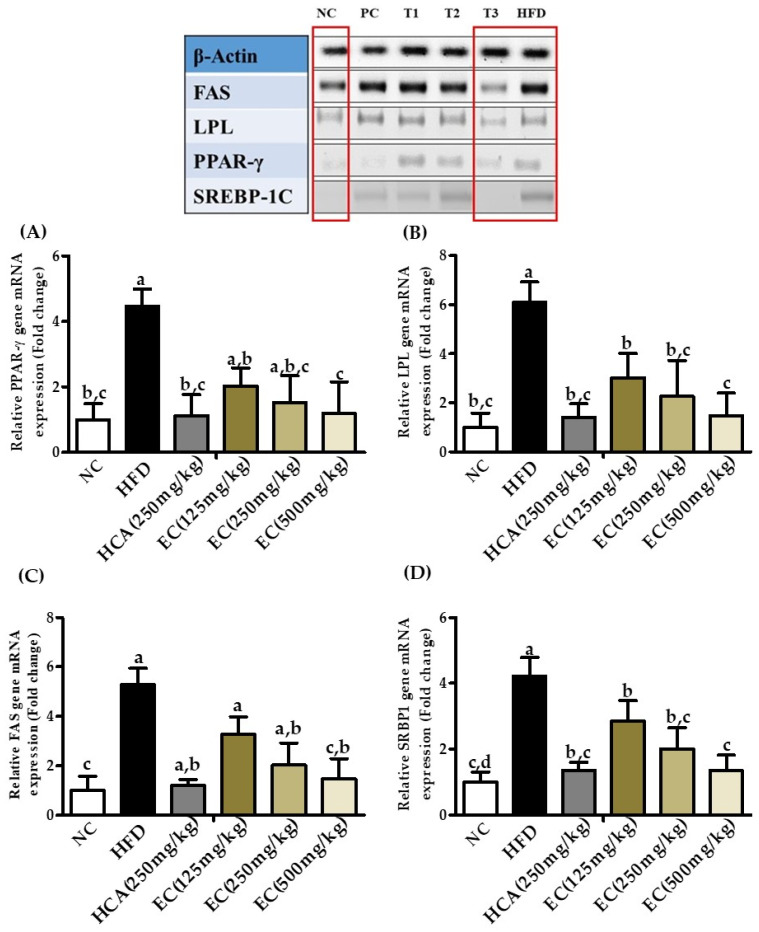
The effects of *Ecklonia cava* extract on adipogenic and lipogenic gene expression. Total mRNA was extracted from different groups of rats, and qualitative gene expression was assessed by polymerase chain reaction (or endpoint PCR). Results obtained from different groups show that the high-fat diet-induced the up-regulation of (**A**) PPAR-γ (peroxisome proliferator-activated receptors gamma), (**B**) LPL (lipoprotein lipase), (**C**) FAS (fatty acid synthase), and (**D**) SREBP-1C genes that can be reduced by the treatment with *E. cava* ethanol (70%) extract. In figure, groups are symbolized as: NC (normal control), PC (positive control, i.e., HCA 250 mg/kg b.w), T1, T2, and T3 shows different doses of EC, i.e., 125, 250, and 500 mg/kg b.w, respectively, and HFD (high-fat diet).

**Figure 12 antioxidants-11-00310-f012:**
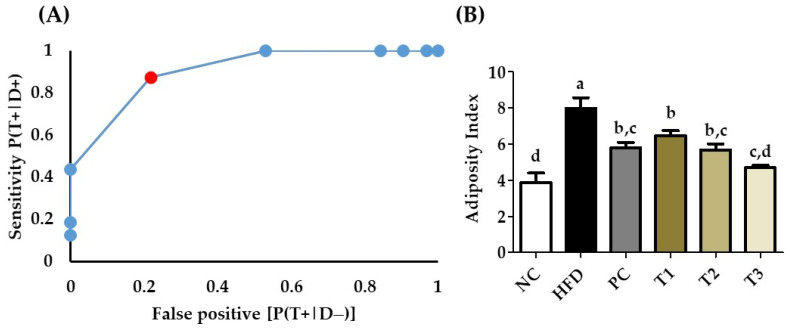
(**A**) Receiver operating characteristic (ROC) curve of body weight index (Red dot represent the cut off value) and area under the curve (ROAUC) and (**B**) adiposity index. Values with different letters present significant differences (*p* < 0.05).

**Figure 13 antioxidants-11-00310-f013:**
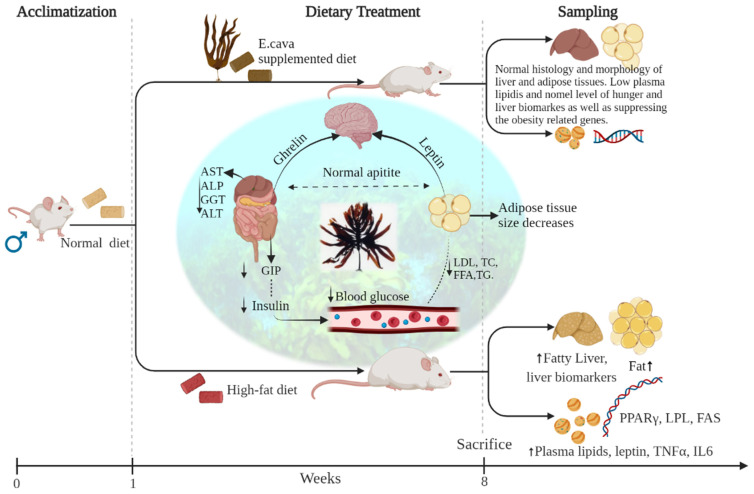
Graphical summary of work represents, in 8 weeks of experiment rats fed with *E. cava* and HFD. After that, a significant difference was observed in the *E. cava*-treated groups and the non-treated groups. The impotent parameters such as a change in body weight, organ histology, and blood plasma biomarkers were compared with the treated and the non-treated groups as well as a normal group of rats.

**Table 1 antioxidants-11-00310-t001:** Daily dose (per kilogram body weight) intake for each group of rats.

Groups		Daily Intake
NC	Normal control	Normal feed
HFD	High-fat diet	Feed with 45% high-fat content (HFD)
PC	Positive control	250 mg/kg b.w HCA * + HFD
T1	Dose 1	125 mg/kg b.w of ECE ** + HFD
T2	Dose 2	250 mg/kg b.w of ECE + HFD
T3	Dose 3	500 mg/kg b.w of ECE + HFD

* HCA, Garcinia extract; ** ECE, *E. cava* ethanol (70%) extract.

**Table 2 antioxidants-11-00310-t002:** Effects of *Ecklonia cava* ethanol (70%) extract on weight gain, % weight gain compared to HFD group, feed intake, feed conversion ratio, and blood glucose (g/dL).

Parameters	Weight Gain (gm/rat)	FCR	Glucose (mg/dL)
NC	388.12 ± 30.3	26.01	102.3 ± 0.54 ***
HFD	509.13 ± 51.7	17.60	139.6 ± 1.85 +++
HCA (250 mg/kg)	431.88 ± 14.6	18.88	103.9 ± 0.95 ***
EC (125 mg/kg)	448 ± 43.7	17.33	105.4 ± 0.75 ***
EC (250 mg/kg)	436.5 ± 23.6	18.01	104.4 ± 0.43 ***
EC (500 mg/kg)	424 ± 14.72	20.08	102 ± 0.61 ***

** FCR was calculated from the total consumption of feed divided by the total body weight gain of 8 rats for 8 weeks; analysis of variance was not used for FCR. Data are presented as means ± SEM (*n* = 8 for each group). * *p* < 0.05, ** *p* < 0.01, *** *p* < 0.001 vs. HFD group, + *p* < 0.05, ++ *p* < 0.01 and +++ *p* < 0.001 vs. normal control group.

## Data Availability

The data of this study will be available at reasonable request.

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
