# Peer review of "Anti-Obesity Effects of Ecklonia cava Extract in High-Fat Diet-Induced Obese Rats"

_antioxidants, 2022, doi:10.3390/antiox11020310_

Round 1
Reviewer 1 Report
This manuscript studies the anti-obesity effects of Ecklonia Cava extract in high-fat diet-induced obese rats. The content of the paper is strongly related to the scope of the journal but it has to be improved considering the below comments.
- Abbreviations need to be defined as they are first used or the authors could define a list of abbreviations. Also a list of nomenclature would be useful considering the equations defined in the paper.
- In the last sentence (Although....) use it was concluded instead of we concluded.
- The authors should consolidate the introduction section in order to emphasize the novelty of the paper and the relevance of your paper for the readership of the journal. In the current form it is not clear what is the advancement achieved in the field and why would this extract be better than ones already reported in the literature.
- All equations need to be numbered and cited in the text. Also, the parameters from the defined equations should be expressed in units according to the SI.
- Based on the obtained result the authors need to compare the performance of Ecklonia Cava extract with other ones reported in the literature and conclude if it is superior or not. Maybe a table with some comparable parameters related to the anti-obesity effects of extracts would be useful.
- I recommend that the authors should consolidate the conclusion section with more numerical results in order to give a more focused view on the study.
Author Response
Dear Reviewer,
Thank you for your consideration of our manuscript. We found the comments quite helpful and we tried to address each point and refined our previous manuscript in accordance with your suggestion. The changes are marked in RED color using “Track Changes” in the revised manuscript. Please find our point-by-point responses to all raised queries in the following:
Kind regards,
Professor, Seung-Chun Park, DVM, PhD
Kyungpook National University.

Reviewer 2 Report
antioxidants-1582681_Anti-obesity Effects of Ecklonia cava Extract in High-fat Diet-induced Obese Rats
The aim of this study is to determine the bodyweight reduction effect of E. cava 70% ethanol extract, as well as an important obesity-related factor, namely, plasma biomarkers, especially related to hunger and liver biomarkers, were evaluated in this study to confirm the antiobesity effect.
This is an original article; the abstract should include the experimental design used.
The introduction is based on relevant articles on the topic. The goal should be more to evaluate not to confirm.
Material and methods should include the approval number of the animal experimentation ethics committee. The design and methodology are correct and it is well explained.
Results: The results are well explained, being the explanatory tables and figures
Discussion: The discussion presents the strengths of the study and uses adequate bibliography on the subject. However, the limitations that it may present should be considered if it´s considered to be able to use it in humans, since the species barrier may be a factor to be taken into account and Therefore, Phase I, II and III clinical trials are necessary before being able to consider its usefulness in humans.
Author Response
Dear Reviewer,
Thank you for your consideration of our manuscript. We found the comments quite helpful and we tried to address each point and refined our previous manuscript in accordance with your suggestion. The changes are marked in RED color using “Track Changes” in the revised manuscript. Please find our point-by-point responses to all raised queries in the following:
Kind regards,
Professor, Seung-Chun Park, DVM, PhD
Kyungpook National University,
